# Immunotherapy for Pulmonary Arterial Hypertension: From the Pathogenesis to Clinical Management

**DOI:** 10.3390/ijms25158427

**Published:** 2024-08-01

**Authors:** Yihan Zhang, Xing Li, Shang Li, Yu Zhou, Tiantai Zhang, Lan Sun

**Affiliations:** State Key Laboratory of Bioactive Substances and Functions of Natural Medicines, Institute of Materia Medica, Chinese Academy of Medical Science and Peking Union Medical College, Beijing 100050, China; zhangyihan@imm.ac.cn (Y.Z.); lixing@imm.ac.cn (X.L.); lishang@imm.ac.cn (S.L.); zhouyu@imm.ac.cn (Y.Z.)

**Keywords:** pulmonary artery hypertension, inflammation, immunotherapy

## Abstract

Pulmonary hypertension (PH) is a progressive cardiovascular disease, which may lead to severe cardiopulmonary dysfunction. As one of the main PH disease groups, pulmonary artery hypertension (PAH) is characterized by pulmonary vascular remodeling and right ventricular dysfunction. Increased pulmonary artery resistance consequently causes right heart failure, which is the major reason for morbidity and mortality in this disease. Although various treatment strategies have been available, the poor clinical prognosis of patients with PAH reminds us that further studies of the pathological mechanism of PAH are still needed. Inflammation has been elucidated as relevant to the initiation and progression of PAH, and plays a crucial and functional role in vascular remodeling. Many immune cells and cytokines have been demonstrated to be involved in the pulmonary vascular lesions in PAH patients, with the activation of downstream signaling pathways related to inflammation. Consistently, this influence has been found to correlate with the progression and clinical outcome of PAH, indicating that immunity and inflammation may have significant potential in PAH therapy. Therefore, we reviewed the pathogenesis of inflammation and immunity in PAH development, focusing on the potential targets and clinical application of anti-inflammatory and immunosuppressive therapy.

## 1. Introduction

Pulmonary artery hypertension (PAH) is a serious cardiopulmonary disease characterized by pulmonary vascular remodeling, causing upward pulmonary artery pressure and ultimately right ventricular failure (RVF). According to the 6th World Symposium on Pulmonary Hypertension (WSPH) in 2018, the definition of pulmonary hypertension (PH) is mean pulmonary arterial pressure (mPAP) higher than 20 mmHg with hemodynamic assessment by right heart catheterization (RHC) [1,2]. Currently, the clinical classification of PH mainly has five groups, including PAH, PH due to left heart disease, PH due to lung diseases or hypoxia, PH due to pulmonary artery obstructions, and PH with unclear or multifactorial mechanisms [2]. PAH is regarded as the first group of PH consisting of idiopathic PAH (IPAH), heritable PAH, drug- and toxin-induced PAH, and PAH associated with connective tissue disease (CTD) as well as other diseases like human immunodeficiency virus (HIV) infection and congenital heart disease [1,3]. The prevalence of PH in line with present estimates is around 1% of the whole population and 10% of people aged more than 65 years. Globally, left heart disease is the leading reason for this disease. The second common cause is lung disease, particularly chronic obstructive pulmonary disease (COPD) [4]. Although effective treatments have been available, the survival rate gradually decreases with the increase in years [5]. The survival rate overall 5-year in PAH patients does not exceed 60% according to epidemiological studies [6]. Enhancing the quality of PAH management and prognosis hinges on precise diagnosis and treatment, with a focus on unraveling the underlying pathological mechanisms of this condition. However, the pathogenesis of PAH is too complex to be exactly understood, and it remains a thought-provoking topic.

Recently, the impression of immunity and inflammation in PAH has gained more attention, interfering with vascular remodeling and contributing to disease progression. Major phenotypes of this disease are vasoconstriction, vascular dysfunction, formation of thrombosis, and inflammation at the site of lesions [7]. It is generally believed that pulmonary vascular dysfunction, as an irreversible tissue change specifically manifests in remodeling and occlusion, is the primary pathological feature for the development of PAH [8]. The increased resistance and hypertension of the pulmonary artery from vascular remodeling is the main reason for PAH pathogenesis, eventually resulting in right heart failure and death of patients [9]. Supportively, researchers have demonstrated that pulmonary artery lesions are in an inflammatory microenvironment, suggesting that inflammatory conditions have a tight relationship to the pathogenesis of PAH (Figure 1A). The aggregation of immune cells and cytokines around the reconstructed pulmonary vessels in PAH corroborated this point [10,11,12]. Equally, the phenotypic changes in pulmonary vascular cells in PAH showed significant correlation with inflammation, including pulmonary artery smooth muscle cells (PASMCs), pulmonary artery endothelial cells (PAECs), and pulmonary artery fibroblasts (PAFs) [12] (Table 1). Moreover, a frequent association between PAH and autoimmune diseases has been demonstrated [13,14]. PAH has been found to initiate more commonly in patients with CTD, especially in systemic sclerosis (SSc) and systemic lupus erythematosus (SLE) [15,16,17]. These findings have revealed that immunity and inflammation probably contribute to the system and progress of PAH.

Over the years, accumulating shreds of evidence has revealed that immunity and inflammation are certainly prominent for the pathogenesis of PAH, supported by multiple preclinical and clinical studies. Although the particular mechanism is still misty, there are multiple potential therapeutic targets from the perspective of immunity and inflammation, which may develop effective treatments to overcome this devastating disease. A better understanding of the role of inflammation and immunity in PAH has both academic and clinical implications and contributions, possibly providing a theoretical basis for the development of immune therapeutic targets. This review summarizes the recent progress regarding immune cells, inflammatory factors, and relevant signaling pathways, elucidating their certain associations and functions in PAH pathogenesis, and concludes with the current immunotherapy, particularly putting emphasis on the potential therapeutic targets and on certain aspects of immunotherapy.

## 2. Immune Cells and Inflammatory Mediators in PAH

A growing number of studies have indicated that a dysfunctional immune system contributes to PAH development. Inflammation is significant among numerous factors for PAH pathogenesis, promoting pulmonary vascular in this disease. Although the exact mechanism of immunity remains unclear, the infiltration of immune cells and inflammatory factors in and around the vasculature has been observed in patients with PAH [7,11]. These are relevant to the formation and dysfunction of pulmonary vasculature, which may become potential targets for the treatment of PAH (Table 2).

### 2.1. Immune Cells

Immune cells might have a substantial impact on PAH, supported by the recruitment and infiltration of immune cells around the remodeled pulmonary vessels present in the lungs of patients with PAH (Figure 1B). Elevated levels of circulating immune cells have been found in PAH patients and animal models, mainly consisting of T lymphocytes, B lymphocytes, macrophages, mast cells, and Dendritic cells [18]. Consistently, the analysis of immune characteristics has demonstrated the correlation and abundance of immune cells, which are significantly different in PAH patients compared to normal subjects [19]. A noticeable study used single-cell RNA sequencing of tissue removed from pulmonary endarterectomy surgery in patients with chronic thromboembolic pulmonary hypertension (CTEPH) to identify multiple cell types including immune cells. It showed that macrophages and CD4+ and CD8+ T cells contribute to chronic inflammation and vascular remodeling in CTEPH through the analysis of phenotypic differences [20]. The notion has been confirmed and gradually elucidated by research in recent years, whereby immune cells and their interactions participate in pulmonary vascular remodeling and inflammatory infiltration in PAH progression (Table 2).

#### 2.1.1. T Lymphocytes

T lymphocyte subsets have been found to play various roles in pulmonary vascular remodeling, mainly including helper T cells (Th cells), cytotoxic T lymphocytes (CTL), and regulatory T cells (Tregs). Th cells and Tregs are differentiated by CD4+ T cells while CTL are from CD8+ T cells, performing the role of immune regulation. There is a significant abnormality in the composition of T cell subsets in the circulating blood and lung tissue of IPAH patients, including the increased CD4+ and CD8+ T cells, suggesting the contribution to the pathogenesis of the disease [21].

The infiltration of CD4+ T cells around pulmonary vessels can be regarded as a characteristic of perivascular inflammation in PAH. In support of this view, recombination–activating gene 1 knockout (Rag1(−/−)) mice, lacking T cells and B cells, were protected from the development of vascular damage induced by monocrotaline (MCT). Further studies confirmed that introducing T cells from control mice to Rag1(−/−) mice can reproduce the vascular injury phenotype when exposed to MCT [22]. Th17 cells, one of the Th cell subsets related to inflammation that produce interleukin (IL)-17, were found to have an increased level in the lung tissue of PAH patients. Consistently, the adoptive transfer of CD4+ T cells or Th17 cells was discovered to restore the pulmonary hypertensive phenotype in Rag1(−/−) mice with chronic hypoxia. The depletion of CD4+ T cells or the treatment of a Th17 cell inhibitor (SR1001) can reverse increased pressure and remodeling responses in chronic hypoxia-induced PH mice [23]. Tregs potentially inhibit PAH progression by exerting negative regulation on T-cell-mediated immune responses. Besides suppressing the proliferation of hypoxia-induced PASMCs, Tregs have been found to have a protective effect on the improvement of hypoxia-induced PAH in mice by regulating the expression of inflammatory cytokines [24]. Dysfunctional Tregs have been observed in various forms of PAH patients, which are associated with the susceptibility of PAH [25,26]. Recent evidence has demonstrated that the reduction in their activity in the inflammatory environment of PAH can lead to the disability of limiting vascular injury [27]. The infusion of Tregs can prevent vascular injury in Treg-deficient rats by anti-inflammatory enhancement and vascular protection, indicating their certain regulatory capacity and therapeutic potential in PAH [26]. In addition, studies have suggested that leptin is detrimentally involved in promoting the pathogenesis of various autoimmune diseases by regulating Tregs [28]. Analogously, secreted by pulmonary endothelial cells, leptin establishes a potential link between endothelial cell disorders and Treg abnormalities, providing a crucial function with the changing of Tregs in patients with PAH [25]. Moreover, the proportion of other T lymphocytes and Tregs can be used as an important biomarker for the prognosis of PAH. The Treg/Th17 ratio has been reported in CTD-PAH patients, the imbalance of which is correlated with the severity and prognosis of this disease [29]. At present, a variety of T cell subtypes have been confirmed to be closely related to the development of PAH, indicating the possibility of targeted T cell therapy.

#### 2.1.2. B Lymphocytes

It has been discovered that B lymphocytes perform a certain role in PAH, involving pulmonary immune response and vascular infiltration in PAH patients and producing antibodies against pulmonary vascular cells [30]. The involvement of B cells in vascular dysfunction has been discovered, based on the fact that the sensitivity to pulmonary vascular remodeling is attenuated in hypoxia- and MCT-induced PAH rats with B cell deficiency [31]. Studies have shown that mice with lung injury are more likely to exhibit PH induced by enhanced activation of B cells and increased B cell receptor signaling, indicating the contribution of adaptive immune activation for inducing the progression of IPAH disease [32]. Importantly, studies have consistently shown that autoimmunity is associated with PAH. Patients with PAH are more likely to develop autoimmune diseases based on clinical statistics. The level of autoantibody is elevated in PAH patients with autoimmune diseases, suggesting the activation and signaling transduction of B lymphocytes in both diseases [33]. It has been discovered that autoantibodies can bind to pulmonary vascular cells like fibroblasts in MCT rats, inducing the generation of proinflammatory phenotypes. Further, the transfer of autoantibody-containing plasmas from MCT rats to naive rats resulted in pulmonary hypertension and vascular remodeling [34]. The above research emphasized the potential impact of B lymphocytes on the correlation between PAH and autoimmunity. Currently, B cell depletion therapy has emerged with positive outcomes in SLE-PAH patients [35]. These studies have implicated the potential of B lymphocytes in the treatment of PAH, especially in combination with autoimmune diseases.

#### 2.1.3. Macrophages

Macrophages are important immune cells that maintain tissue homeostasis and the stability of the immune system, making a crucial impression on the disease process of PAH. As the main infiltrating inflammatory cells around the lesions of PAH, macrophages contribute to pulmonary vascular remodeling by secreting multifarious chemokines and certain growth factors [36]. The polarization phenotypes of macrophages are mainly divided into M1 and M2 types, performing different effects in inflammatory responses. M2 macrophages secrete amounts of anti-inflammatory cytokines while M1 macrophages produce pro-inflammatory factors [37]. However, they are not static due to their plasticity in response to environmental influence, in accordance with the phenomenon that the phenotype of pulmonary macrophages varies over time in PAH models [38]. The inference has been reached from studies that the imbalance of the M1/M2 ratio might be relevant to the aggravation of PAH, by changing the inflammatory environment around pulmonary vessels [39]. It has been found that donepezil can inhibit inflammatory responses and macrophage activation in rat models of PAH, reducing the proliferation of PASMC to alleviate PAH progression [40].

The occurrence of perivascular inflammatory response in PAH is closely related to the recruitment of macrophages, proving their promotion for pulmonary vascular remodeling [36]. It has been confirmed that macrophages can promote the proliferation of PASMC, exacerbating vascular lesions in PAH patients, between which C-C chemokine receptor (CCR) 2 and CCR5 play an important role in the collaboration [41]. Matrix metalloproteinase (MMP)-10 from macrophages that infiltrate the lungs was found to have increased expression in MCT- and hypoxia-induced PAH rats, which also exacerbate PASMC proliferation and migration [42]. It has been reported that inhibition of runt-related transcription factor 1 (RUNX1) can reduce the recruitment and activation of pulmonary macrophages in SU5416/hypoxia-induced PH rats, further inhibiting the development of disease [43]. Additionally, the increased number of right ventricular (RV) macrophages and the activation of nucleotide-binding oligomerization domain-like receptor protein 3 (NLRP3) inflammasomes have been observed in PAH rat models and patients, suggesting that macrophages may induce RV inflammation leading to RVF in PAH [44]. Concluded from the above, the recruitment of macrophages in pulmonary vascular lesions and the presence of related cytokines consequentially participate in the progression of PAH. Targeted therapy of macrophages may provide clinical significance for improving the deterioration of PAH.

#### 2.1.4. Mast Cells

Mast cells regulate inflammatory activities primarily through degranulation, releasing cytoplasmic granules to secrete abundant proteases, lipid mediators, cytokines, and chemokines after stimulation. Mast cells have been histopathologically observed to accumulate around pulmonary vessels and participate in pulmonary vascular remodeling in PAH models and patients. It has been found that the levels of the above molecules produced by mast cell degranulation are elevated in PAH patients [45,46]. Consistently, analyzing the dynamic changes in mast cells of the MCT-treated rats showed that over time mast cells excessively infiltrated and degranulated in lung tissue but not RV, participating in pulmonary vascular lesions by releasing cytokines such as tryptase [46]. Previous studies have shown that the extent of pulmonary vascular remodeling and PAH are significantly reduced in left heart disease rats with mast cell stabilizer ketotifen and mast cell-deficient rats [47]. It has been elucidated that total tryptase and urine leukotriene E4, as the biomarkers of the activation of mast cells, decreased after mast cell inhibitor blockade. And the treatment also led to the reduction in vascular endothelial growth factor (VEGF) and proangiogenic CD34+CD133+ progenitor cell levels, implying the contribution of mast cells to pulmonary vascular dysfunction [48]. Moreover, growth factor receptor kinase c-KIT presenting on mast cells mediates signaling pathways associated with vascular remodeling, supported by the high expression levels of c-KIT in lung tissues of IPAH patients [49]. Kinase inhibitors like imatinib targeting c-KIT, have been elucidated to benefit patients with PAH [50].

According to recent research, the mediators released by the degranulation of mast cells can interact with vascular cells to promote remodeling. A significant increase in chymase-positive mast cells was found in the lungs of IPAH patients, positively correlating with the total number of mast cells [45]. Mast cell chymase indirectly alleviates pulmonary vasoconstriction through stimulating the production of angiotensin II (AngII), endothelin-1 (ET-1), and MMP [48,51]. With the treatment of the chymase inhibitor, rats with flow-associated PAH created by MCT and an aortocaval shunt showed attenuated pulmonary vascular remodeling, implying the therapeutic potential of targeting chymotrypsin [52]. Besides the protease from degranulation, lipid mediators produced by mast cell activation like prostaglandin D2, leukotriene B4, leukotriene C4, and platelet-activating factor have gradually attracted attention and verified function to PH development [53]. Recently, ω-3 fatty acid-derived epoxides, derived from tryptase-positive mast cells by platelet-activating factor acetylhydrolase, were found to regulate pulmonary vascular remodeling by inhibiting the abnormal activation of pulmonary fibroblasts through transforming growth factor-β (TGF-β) signaling. In vivo administration of ω-3 fatty acid-derived epoxides can attenuate the exacerbation of PH in animal models, showing the potential for inhibiting pulmonary vascular remodeling and treating PAH [54]. The activation and accumulation of mast cells around diseased pulmonary vessels propose a suggestion for their involvement and promotion in the pathogenesis of PAH. Targeting the mediators generated by mast cells, including diverse proteases and lipid mediators, has shown promising therapeutic potential for ameliorating disease progression.

#### 2.1.5. Dendritic Cells

Dendritic cells (DCs), with the essential function of antigen presentation, produce multiple cytokines after activation to regulate T cells and B cells. Various phenotypically and functionally distinct subsets have been identified, including type 1 cDCs (cDC1s), type 2 cDCs (cDC2s), monocyte-derived DCs (MoDCs), and plasmacytoid DCs (pDCs) [55]. Previous studies have found that DCs accumulate around remodeling pulmonary vessels and migrate to pulmonary lymph nodes and tertiary lymphoid organs (TLOs) in patients with PAH [56]. Consistently, TLOs were found in the vicinity of pulmonary vessels in IPAH and CTD-PAH patients, with the recruitment of multiple subsets of DCs [55]. These suggested that DC cells together with other immune cells might collectively affect pulmonary vascular function. Especially, the interaction with DCs and T lymphocytes in pulmonary vascular lesions is noticed, which may promote the progression of PAH. In recent studies, co-culture of MoDCs with T lymphocytes in IPAH patients upregulated the activation and proliferation of CD4+T lymphocytes, and enhanced the polarization of specific Th17 cells [57]. Consistent with that, co-localization of DCs and CD8+T cells was observed in the lung tissues of IPAH patients, indicating a close relationship between DCs and T cells around the vascular and parenchymatous tissues of the lung [58]. Therefore, targeted therapy for DCs, by influencing the immune environment around pulmonary vasculature, may have the possibility of contributing to the improvement of PAH.

#### 2.1.6. Neutrophils

Neutrophils participate in immune responses and support the innate and adaptive immune systems by performing a regulatory role [59]. It is suggested that the recruitment of neutrophils at the lesions of pulmonary perivascular inflammation in PAH patients indicates the effect on vascular remodeling progression [60]. In particular, neutrophil elastase (NE) was found to correlate with the pathogenesis of PAH [61]. The circulating NE level has increased in patients with different subtypes of PAH, relevant to poor clinical prognosis in terms of disease severity [62].

It is well-known that the role of neutrophils in immune response is phagocytosis, degranulation, and the release of neutrophil extracellular traps (NETs). NETs are extracellular reticular structures released by neutrophils that immobilize and even destroy pathogens, confirmed to have a pathological correlation with PAH [63]. In recent studies of patients with IPAH and CTEPH, the presence of neutrophils and neutrophil extracellular trap formation (NETosis) around pulmonary vascular lesions has been elucidated. And the biomarkers of NETosis in plasma were increased in the above patients, suggesting the involvement of activated neutrophils in PAH. As further studies have shown, NETs contribute to proinflammatory and proangiogenic responses in PAECs by the activation of myeloperoxidase/hydrogen peroxide (H_2_O_2_)-mediated toll-like receptor (TLR) 4/nuclear factor-kappa B (Nf-κB) signaling [64]. Additionally, increased neutrophil production and released NE in PH patients are correlated with enhanced NETs by proteomics and transcriptomics analysis [65]. Consistent with expectations, NETs exactly regulate pulmonary vascular homeostasis in PAH models, contributing to endothelial angiogenesis and vasoconstriction.

Recently, neutrophil elastase inhibitors have been developed to treat heart and lung-related diseases [66]. For instance, an endogenously produced NE inhibitor called elafin is on the way to early therapeutic application. Previous findings have indicated that elafin can inhabit NE activity and reverse PH symptoms in SU5416/hypoxia rat models, and promote angiogenesis by increasing bone morphogenetic protein receptor 2 (BMPR2) signaling in patient PAECs [67]. In recent trials of elafin therapy for PAH, it was verified that elafin can improve PAEC homeostasis and mitigate PAH [62]. The conclusion from these studies is that neutrophils have a non-negligible influence on the occurrence and development of PAH, especially the significance of NE and NETs. Inhibition of NE has revealed promising benefits in treating PAH, suggesting a certain clinical potential for targeting neutrophils.

### 2.2. Cytokines

There are abundant cytokines around pulmonary vessels regulating the pathological progression of PAH, causing pulmonary vascular dysfunction and increasing the severity of the disease (Table 2) [10,12]. These inflammatory factors are at elevated levels in PAH models and patients, which may be considered biomarkers for clinical diagnosis and outcome of this disease [68,69]. Clinical investigations have demonstrated the dysregulation of multiple inflammatory mediators in various forms of PAH, such as IL-1, IL-6, TGF-β, and CCR2 (Figure 2) [12,18,70,71]. Targeting the regulation of inflammatory cytokines and their signaling transduction or controlling upstream pathways associated with cytokine release may prevent the progress of this disease. Importantly, the regulatory effect of these cytokines and chemokines offers a promising avenue for targeted intervention, thereby potentially innovating the diagnosis and treatment for PAH patients.

#### 2.2.1. TGF-β

Transforming growth factor-β, as a multifunctional cytokine belonging to the transforming growth factor superfamily, is noticeably responsible for the progression of PAH. With the ability to regulate homeostasis fundamental processes, TGF-β influences cell proliferation, differentiation, apoptosis, migration, adhesion, and inflammation [72]. In recent years, the conspicuous function of TGF-β signaling in PAH pathogenesis has been widely concerned and gradually elucidated. TGF-β pathway transmits signals through small mothers against decapentaplegic (Smad) and other factors, including phosphatidylinositol 3 kinase (PI3K)—protein kinase B (AKT), extracellular signal-regulated kinase (ERK), p38 mitogen-activated protein kinase (MAPK), or c-Jun N-terminal kinases (JNK) (Figure 2) [73,74]. Accumulating studies have elucidated the critical effect of bone morphogenetic protein (BMP)/TGF-β signaling pathway in PAH progression. The presence of BMPR2 mutation or reduction is commonly present in patients with various types of PAH, leading to the aberrant activation of TGF-β signaling, which contributes to the disease pathogenesis [74,75].

The disequilibrium of BMP/TGF-β signaling is associated with endothelial dysfunction, vascular remodeling, inflammation, and angiogenesis disorders in PAH [73,76]. TGF-β was found to have increased levels in serum and lung tissue in patients with iPAH compared to controls, which proposes a clue regarding the involvement of TGF-β in the PAH process [77]. Consistent with that, the secretion of TGF-β was observed to increase in PASMCs of PAH patients. Further investigation demonstrated the promotion of TGF-β to PASMC proliferation in PAH, supported by the fact that the PAH-conditioned medium significantly increased the phosphorylation of Smad2, Smad3, and Smad1/5 and the expression of AKT, ERK1/2, and p38 MAPK in non-diseased PASMCs. In support of this view, the inhibitory anti-TGF-β antibody can significantly inhibit pSmad3, pERK1/2, and the growth of PASMCs in PAH [78]. Moreover, studies revealed that TGF-β1 may protect PASMCs from apoptosis by activating the PI3K/AKT pathway, showing the promotion of growth and inhibition of death in PASMCs [79]. In addition, verification that TGF-β1 released from platelets participating in PAH progression has been recognized. TGF-β1 increased the expression of pyruvate kinase muscle 2 via mammalian target of rapamycin (mTOR)/cellular-myelocytomatosis viral oncogene (c-Myc)/polypyrimidine tract binding protein 1 (PTBP-1)/heterogeneous nuclear ribonucleoprotein A-1 (hnRNPA-1) signaling, which enhances aerobic glycolysis of PASMCs after platelet activation. Further research has indicated that platelet TGF-β1 deficiency mice can improve ventricular systolic pressure (RVSP) and attenuate pulmonary vascular remodeling from SU5416/hypoxia-induced PAH [80]. These findings prove the significance of TGF-β signaling in pulmonary vascular dysregulation, suggesting the discovery of clinically meaningful therapeutic targets. Inhibitors targeting TGF-β1 signaling may be an efficient approach to improve the disease situation of PAH. For instance, it was confirmed that a prostacyclin analog called treprostinil can inhibit the TGF-β pathway by suppressing Smad3 phosphorylation in MCT-induced PAH rats with increased TGF-β signaling [81]. Recently, the clinical trials of a high-profile drug called Sotatercept targeting TGF-β showed promise for treating PAH, matching TGF-β ligands to inhibit vascular remodeling and ameliorate disease progression [82]. In general, it is promising to target TGF-β and proteins relevant to the signaling, which has significant therapeutic potential for clinical application.

#### 2.2.2. IL-6

IL-6 is a cytokine of great importance with diverse biological influences, performing a crucial role in immune and inflammatory responses. Over the years, the function of IL-6 in PAH pathogenesis has been increasingly discovered and confirmed. Accumulating evidence has demonstrated that IL-6 levels are elevated in patients with different forms of PAH, relevant to clinical deterioration and poor prognosis of disease, to a large extent [70,83,84]. In particular, researchers found that the serum IL-6 level was higher in CTD-PAH than in other types of PAH, with a prominent correlation with the mortality of patients [84,85]. Recent studies have confirmed the significant elevation of IL-6 expression in PASMCs from PAH patients with different types, implying the involvement of IL-6 in pulmonary vascular remodeling [81]. Consistent with this view, IL-6 has been found to promote the proliferative activity of PASMCs [86,87]. It has been discovered that activation of the mTOR/ribosomal protein S6 kinase1 (S6K1) pathway is responsible for the proliferation of PASMCs via paracrine IL-6 induced by senescence PASMCs [86]. In addition, studies have revealed the correlation showing that RV dysfunction exhibited a more obvious severity with higher IL-6 levels in PAH patients, supporting that IL-6 performs a pathogenic role in different pathological mechanisms of PAH [88]. Furthermore, IL-6 importantly correlates with immunoglobulin secretion and autoimmune antibody production, stimulating the differentiation of B cells into plasma cells [31,51].

The appearance and importance of the IL-6 signaling pathway have been emphasized over the past decades. IL-6 can bind to both membrane-bound and soluble IL-6R (sIL6R) to form a complex that initiates downstream signaling, including Janus kinase (JAK)/signal transducers and activators of transcription (STAT) 3 and other signaling pathways such as PI3K/AKT and MAPK/ERK (Figure 2) [89]. This IL-6/IL-6R signaling has been demonstrated to be present in PAH. It has been shown that circulating IL-6 and sIL6R and lung levels of membrane-bound IL6R were found to increase in MCT rat models [90]. Consistently, serum concentrations of IL-6 and sIL6R were observed at higher levels in patients with PAH than in control subjects [91]. In addition, the membrane-bound IL-6R was ectopically upregulated in PASMCs of IPAH and PAH patient models, which also present a critical role in PASMC accumulation in vitro and in vivo [92]. These findings have elucidated that IL-6 signaling and trans-signaling are manifested and enhanced in PAH, aiming at these pathways and molecules may have the potential to target this disease. In recent clinical studies, it has been identified that treatments with IL-6R antagonists can attenuate pulmonary vascular remodeling that prevents or reverses the development of PAH in MCT and SU5416/hypoxia rat models [81,90]. IL-6 is gradually recognized as a vital factor in the progression of PAH. Importantly, the inhibition of IL-6 could be of clinical significance as a pharmacological strategy in PAH, showing positive anticipation for the time to come.

#### 2.2.3. IL-1

IL-1 family cytokines are crucial signaling molecules in the innate and adaptive immune systems that mediate inflammatory responses to different stimulations. The IL-1 family is composed of three ligands including IL-1α, IL-1β, and the IL-1 receptor antagonist (IL-1Ra), all of which are related to PAH on account of the elevated levels of IL-1 in this disease [70,92]. IL-1R consists of two distinct forms, including IL-1RI and IL-1 accessory protein (IL-1RAcP). IL-1 can bind to IL-1RI and IL-1RAcP to form a complex and activate downstream signaling transduction, such as NF-κB, activator protein-1 (AP-1), JNK, and MAPK pathways (Figure 2) [93]. Notably, IL-1RI is highly homologous to cytoplasmic domains with TLR, both affecting antigen recognition and immune function [92].

As an important proinflammatory cytokine, IL-1β is primarily involved in a variety of inflammatory responses and cellular activities, including cell proliferation, differentiation, and apoptosis. A growing body of research has shown that IL-1β presents an elevated serum level in PAH, relevant to the prognosis of the patients [70,94]. The proliferative activity in human aortic smooth muscle cells (HASMCs) was found to be enhanced by the presence of IL-1β, indicating that IL-1β may contribute to vascular remodeling [95]. Similarly, it has been confirmed that monocyte-secreted IL-1β and IL-18 can induce amplified proliferation of vascular smooth muscle cells (VSMCs) [96]. IL-1β may directly or indirectly affect the pulmonary artery vasoconstriction and remodeling, influencing the proliferation and inflammation of pulmonary vascular cells in PAH. It has been discovered that IL-1β produced from pulmonary macrophages can indirectly increase PASMC proliferation and mediate inflammatory infiltration. The activation of the IL-1β signaling pathway and NLRP3 inflammasome is promoted by caspase-8 during the macrophage-associated inflammation in PAH [97]. With reduced BMPR2 signaling, IL-1β was suggested to induce exaggerated activation of inflammatory response in PASMCs and mice, demonstrating a connection between IL-1β and BMPR2 in the PAH pathogenesis [94]. Moreover, studies found that IL-1β can promote PASMC proliferation and vascular dysregulation via the IL-1RI/myeloid differentiation primary response protein 88 (MyD88) pathway in PAH. As a crucial adaptor protein transducing signal in innate immunity, MyD88 is involved in IL-1RI and TLR signaling, inducing the synthesis of IL-1, IL-6, and tumor necrosis factor -α (TNF-α) through activation of Nf-κB [98,99]. In terms of support, the expression of IL-1RI and MyD88 was significantly increased in the lung tissues of IPAH patients and PASMCs of hypoxic-induced PAH mice. Further studies suggested that the mice with MyD88 gene deficiency, IL-1RI gene deficiency, or anakinra-induced IL-1RI inhibition similarly showed the improvement of hypoxic PAH. Additionally, the reduction in perivascular macrophage accumulation was found in hypoxia-induced PH in the mice described above [98]. Recent clinical research reported the safety and feasibility of IL-1R antagonist (anakinra) in patients with PAH and RV failure, suggesting that IL-1 blockade may have the potential for treatment [100]. According to the above, based on the impact of IL-1 in PAH, therapeutic agents targeting IL-1, particularly IL-β and IL-1R, may become an effective strategy in terms of improving disease deterioration.

#### 2.2.4. MCP-1

Monocyte chemoattractant protein-1 (MCP-1), also called C-C chemokine ligand (CCL) 2, is a pro-inflammatory chemokine with chemotactic activity for monocytes and basophils. It is one of the crucial chemokines that regulate monocyte/macrophage migration and infiltration. Previous clinical studies have shown that circulating MCP-1 levels are significantly increased in patients with IPAH and negatively correlated with disease duration, suggesting the involvement of MCP-1 in the PAH pathological process (Figure 2) [101]. Consistent with that, MCP-1 is overexpressed in both plasma and lung tissues of IPAH patients, also exhibiting chemotactic activity to PAECs and PASMCs with elevated levels of MCP-1 [102]. These facts support the point of view that abnormal phenotypic changes in pulmonary vessels are correlated with MCP-1. Additionally, 15-Lipoxygenase (15-LO)/15-hydroxyeicosatetraenoic acid (15-HETE) signaling can promote coagulation and platelet activation in the pulmonary vascular thrombotic lesions of PAH models, which was suppressed and reversed by MCP-1 inhibition in hypoxia-induced rats [103]. Recently, studies have found that MCP-1 participates in myocardial hypoxia through its regulation of miR-146b, which may have been correlated with heart failure caused by PAH. It has been observed that hypoxia downregulated miR-146b and induced the expression of tumor necrosis factor receptor-associated factor 6 (TRAF6), IL-6, and MCP-1 in both the right and left ventricles of mice with induced alveolar hypoxia, suggesting that the miR-146b/TRAF6-IL-6/MCP-1 axis may contribute to ventricular inflammation and dysfunction [104]. Based on its role in the regulation of inflammation and vascular remodeling in PAH, MCP-1 possesses the potential to be a therapeutic target. In recent studies, due to the inhibition of the CCL2/CCR2 inflammatory pathway in the lung tissue of MCT-indued PAH rats, crocin has shown the clinical capacity to improve pulmonary vascular dysfunction and prevent PAH progress [105]. As a chemokine that is present and alters pulmonary vascular homeostasis, MCP-1 has shown adverse effects in PAH. Based on these studies, inhibition of MCP-1 may be a targeted strategy with clinical therapeutic implications.

## 3. Inflammation-Related Signaling Pathways in PAH

Based on studies of immune cells and inflammatory mediators in PAH, the existence of potential therapeutic targets associated with inflammation of clinical significance has been gradually exposed. In particular, the signaling pathways associated with inflammation like JAK-STAT, Nf-κB, and TLR have been revealed to provide an inspiring avenue for the clinical diagnosis and treatment of this disease [106,107,108,109].

### 3.1. JAK-STAT

The JAK-STAT pathway is regarded as a pivotal inflammatory signaling pathway, contributing to the regulation of the immune system and participating in the procession of cells like development, differentiation, proliferation, and apoptosis [107]. Recent studies have demonstrated that the activation of the JAK-STAT pathway in pulmonary arteries of patients with different PH types is excessive. Consistent with this, the expression of JAK and STAT has been observed excessive in clinical patients and preclinical animal models of PH. Dysregulation of certain factors in the pathogenesis of PH contributes to the activation of the JAK-STAT pathway, which is associated with vascular remodeling, cell proliferation and apoptosis resistance, and inflammation progression (Figure 3) [110].

JAK-STAT signaling is a hot field for immune-related diseases with clinical significance. Targeting the JAK-STAT pathway is a highly promising treatment option for pulmonary hypertension, which has been confirmed to be correlative with immunity and inflammation, as demonstrated by recent studies. Previous evidence indicated that certain isoforms of JAK and STAT have elevated mRNA and protein levels in hypoxic human PASMCs [111]. Recent studies in patients with idiopathic pulmonary fibrosis (IPF) and PH have also shown increased expression of JAK2 and STAT3 in pulmonary arteries. It can be inferred that JAK2 is involved in pulmonary vascular remodeling and the formation of pulmonary hypertension, supported by rats treated with intratracheal bleomycin and the PASMCs and PAECs in pulmonary arteries of IPF patients [112]. More pertinently, researchers have elucidated the feasibility of JAK2 as a therapeutic target in patients with PAH. This has been found that the activation of JAK2/STAT3 in hypoxic-induced PAH mouse models and human PASMCs. Furthermore, the depletion of JAK attenuated pulmonary vascular remodeling and PASMC proliferation, protecting the mice with SMC-specific JAK2 deficiency under chronic hypoxic conditions from the disorders of pulmonary pressure and RV function. Consistently, JAK2 inhibition can restrain the proliferation of human PASMCs when the inhibitor is applied [113]. Based on the above, JAK inhibitors have shown great promise for clinical treatments of PH patients, with many ongoing studies in progress to confirm their effectiveness.

Ruxolitinib is a JAK1 and JAK2 inhibitor, especially for treating myelofibrosis (MF), which has recently been explored for the potential and possibility of being a targeting PH therapy for the known but easily neglected complications of MF [114]. It has been previously demonstrated that the treatment of ruxolitinib for MF-associated PH patients has shown a reduction in biomarkers and certain cytokines relevant to PH, contributing to the improvement in disease-related symptoms [115]. The JAK signaling activation might be mediated by imbalanced cytokines, causing the progression of PH. On the strength of the preclinical studies in animals and patients with PH, JAK inhibitors represent the achievable clinical remedy. A recent study in rats with CTEPH showed that oral treatment of ruxolitinib dose-dependently attenuated pulmonary vascular remodeling and reversed the generation of this disease [116]. Further studies elucidated the effectiveness of ruxolitinib in PAH patients, both in vivo and in vitro experiments provided the evidence. As the research demonstrated, ruxolitinib has the ability to inhibit the JAK2-STAT3 signaling pathway to reduce and reverse pulmonary vascular remodeling. It has been confirmed that ruxolitinib can restrain the proliferation and migration of human PASMCs stimulated by IL-6 in IPAH patients and facilitate the improvement of cardiopulmonary function in MCT- or hypoxia-induced PH rat models [117]. Above all, the studies have shown that JAK inhibitors, represented by ruxolitinib, have the potential of dose-related treating this disease, blazing a way in the clinical immunotherapy of PH. There are other considerable drugs targeting the JAK-STAT signaling pathway showing efficacy in preclinical studies, painting a promising therapeutic prospect for the status of PH disease (Table 3).

### 3.2. Nf-κB

As is proven to all, the Nf-κB signaling pathway participates in multifarious processes of biological organisms, contributing to regulating diverse vital activities of cells like immunity and other responses. Regarded as a transcriptional factor involved in regulating multiple genes, Nf-κB plays a critical role in cellular inflammation, the dysregulation of which will result in diseases related to the immune system. A growing amount of evidence has shown that Nf-κB signaling has the possibility to make a correlation between pathological dysfunction and inflammation in PH.

Previous research has explored the pathophysiological roles of the Nf-κB pathway in vitro and in vivo experiments, implicating the possibility of a critical impact on Nf-κB inhibitors targeting PH disease. Recently, a single-cell study published the outcomes that Nf-κB signaling was broadly upregulated in different cell types from Sugen-hypoxia- and MCT-induced PAH rats [108]. The preclinical effectiveness and clinical potential of the Nf-κB inhibitor (N-(3,5-Bis-trifluoromethyl-phenyl)-5-chloro-2-hydroxy-benzamide, IMD-0354) recently have been elucidated in PAH. It can improve physical disease symptoms and prevent pulmonary vascular remodeling in MCT-induced PAH rat models, which also suppressed proliferation and induced apoptosis in PASMCs [119]. Another study for CTEPH demonstrated the activation of Nf-κB signaling and related genes in C-reactive protein (CRP)-induced PAECs, improved by the Nf-κB inhibitor pyrrolidine-dithio-carbamate ammonium (PDTC) [120]. Additionally, the dipeptidyl peptidase-4 (DDP4) inhibitor sitagliptin can suppress the activation of the mTORC1/Nf-κB/DDP4 pathway, attenuating the vascular dysfunction in lung tissues of hypoxia-induced PH rats [121]. Accordingly, the Nf-κB pathway has been elucidated as relevant to the pathogenesis of PH in available preclinical studies, indicating the therapeutic potential of targeting Nf-κB signaling.

### 3.3. TLR

Represented as one of the active and vital members of inflammatory responses, TLRs are type I transmembrane glycoprotein receptors that play a significant role in mediating innate immunity [122]. In the immunological stage, the recognition of TLR for congenital immunity activates the body to make immune responses, contributing to a bridge between specific and non-specific immunity. Based on previous research, TLR and its associated signaling pathways are implicated in the impact of complications in multifarious vasculopathy including endothelial dysfunction, vascular inflammation, arterial hypertension, and atherosclerosis [123]. TLR activates downstream signaling like the MAPK and Nf-κB pathways for the induction of proinflammatory cytokines [122,124]. The compact connection between TLR and immune response is in accordance with vascular inflammation and dysregulation, based on the general distribution of TLRs in cells of blood vessels. A recent bioinformatic analysis elucidated the potential of the dysregulated TLR pathway in the pathogenesis of MCT-induced PAH, including inflammatory immune response and pulmonary vascular dysfunction, proposing new insights into the immune mechanism of PAH [125].

Given the differences that exist among the TLR family, the relation and contribution of different TLRs in aggravating vascular inflammation and endothelial dysfunction of PH progression need further investigation. It has been evidenced that the TLR4 deficient mice with hypoxia-induced PAH showed resistance to PH compared to the mice with TLR4 intact, suggesting that TLR4 may contribute to the susceptibility to PH by intensifying pulmonary vascular inflammation [126]. Another study focused on TLR7/8 showed that topical application of the TLR7/8 agonist resiquimod (R848) in SU5416 rats exhibited severe symptoms of PH and autoimmune disorders, implicating the immunopathological link of autoimmune vasculopathy [127]. Moreover, the participation of TLR9 has been demonstrated in the pathogenesis of PAH through activating the NF-κB/IL-6 pathway, confirmed in PH rats with MCT-induced PAH. Selective (E6446) and nonselective TLR9 inhibitor (chloroquine) TLR9 inhibitors showed the ability to attenuate, reverse, and prevent the process of disease in rat models with PH, improving pulmonary vascular dysfunction and prolonging survival [128]. Besides the inhibitors, recent studies suggest that TLR3, associated with vascular protection, emerges as a targeting potential for PH. With the decreased expression of TLR3 in lung tissues and endothelial cells of PAH patients, TLR3 deficiency might contribute to PAH for endothelial apoptosis and pulmonary vascular remolding. In addition, a TLR3 agonist (polyinosinic/polycytidylic acid (Poly[I:C])) reduced PAH progression in the lungs of chronic hypoxia and SU5416/hypoxia rat models [109]. In conclusion, TLR has been demonstrated to be correlated with pulmonary vascular abnormalities in preclinical studies, suggesting that targeting these proteins might conduct the clinical possibility of PH treatment.

## 4. Anti-Inflammatory and Immunosuppressive PAH Therapy

As a life-threatening disease with poor clinical manifestations, effective therapeutic drugs are urgently needed in patients with PAH. Most of the drugs approved to treat PAH are vasoconstriction inhibitors. Treatments targeting classical pathways of PAH pathogenesis, including the ET-1, prostacyclin, and nitric oxide (NO) pathways, have been widely utilized in clinical practice (Figure 4) [93,106]. However, these treatments are just used for inhibiting pulmonary vasoconstriction to reduce pulmonary vascular resistance. Vasodilation with drugs relieves patients’ disease symptoms, but it is unable to fundamentally overcome vascular remodeling.

In recent years, the role of inflammation and immunity in PAH has attracted increasing attention. Accumulating evidence demonstrated the involvement and influence of inflammation in the pathological mechanism of PAH, implicating that it may inspire the generation and application of new therapeutic drugs [7,18,93]. Currently, certain immune-related medications have been demonstrated to improve or even reverse the development of PAH, although these types of drugs are not yet available on the market (Table 4). Immunotherapy may have clinical potential in the possibility of applying alone or in combination with PAH treatment, showing a promising prospect to benefit patients (Figure 4).

### 4.1. Tocilizumab

As one of the IL-6 receptor antagonists, tocilizumab achieves immune modulation to have the potential of ameliorating the progression of PAH by blocking the IL-6 signaling pathway. This humanized monoclonal antibody that recognizes IL6R has influence on disrupting both classical and trans-signaling of IL-6 [132]. Tocilizumab can inhibit IL-6 by binding to its specific receptors, which contributes to preventing the pro-inflammatory and fibrotic effects [133]. Recently, an open-label, single-arm, phase II study (NCT02676947) of intravenous tocilizumab with 8 mg/kg over 6 months in patients with group 1 PAH, aiming to assess the safety and efficacy, was completed. However, tocilizumab failed to deliver the expected result, which led to no significant effect on PAH including IPAH, hypoxic PAH, and CTD-PAH [130].

### 4.2. Rituximab

Recent studies have reported that rituximab, as an anti-CD20 chimeric monoclonal antibody that targets the B lymphocyte CD20 protein, has emerged with effective potential in treating CTD-PAH and SLE-PAH patients [134,135]. Based on the significant role of inflammation in CTD-PAH pathogenesis, clinical studies have shown that immunosuppressive therapy shows a beneficial effect on CTD-PAH patients, especially in the SLE-PAH among CTD subtypes. Interestingly, one randomized controlled trial evaluated the efficacy of rituximab in CTD-PAH patients, which indicated the potential of rituximab in clinical treatment [136]. B lymphocyte depletion therapy may probably be of benefit in the treatment of SSc-PAH. On account of a multicenter, double-blinded, randomized, placebo-controlled, proof-of-concept clinical trial (NCT01086540), rituximab was evaluated as adjuvant therapy in patients with SSc-PAH. After receiving two infusions of 1000 mg rituximab or placebo administered 2 weeks apart for 24–48 weeks, the safe and well-tolerated effect of rituximab treatment was mostly observed in patients. Rituximab significantly improved the 6 min walk distance (6MWD) at 24 weeks considering the data of 48 weeks, showing a beneficial effect on patients with SSc-PAH in clinical implications [35]. According to the current situation, further studies on rituximab and other immunosuppressants may be valuable for the clinical treatment of PAH.

### 4.3. Sotatercept

Sotatercept is a fusion protein that is composed of the human activin receptor type IIA (ActRIIA) linked with human IgG, binding and traping the ligands of ActRIIA, other activins, and growth differentiation factors in the TGF-β superfamily [82]. This first-in-class drug aims at rebalancing the dysregulation of TGF-β signaling to inhibit cellular proliferation and relieve vascular inflammation in the treatment of PAH. Based on the positive effect in preclinical animal studies, sotatercept has entered the clinical trial stage, consistently showing impressive and encouraging results.

Recently, sotatercept completed the phase III STELLAR trial (NCT03496207) with therapeutic benefits in PAH patients, achieving the potential for clinical treatment [137]. As for the previous phase II trials accessing the longer-term safety and efficacy of sotatercept, PAH patients received the treatment of subcutaneous sotatercept every 3 weeks at a dose of 0.3 or 0.7 mg/kg for 24 weeks (NCT03738150) and further for 18 to 24 months (NCT03496207) [138,139]. The above trials suggested that sotatercept can significantly reduce pulmonary vascular resistance and improve exercise capacity measured by 6MWD both in the short and long term with acceptable safety and durability. Encouragingly, sotatercept treatment significantly improved both pulmonary hemodynamics and right heart function in patients with PAH, achieving multiple efficacy endpoints in the STELLAR trial. With the therapeutic effect resulting in clinical advantage, sotatercept markedly lowered the disease deterioration and death risk in PAH patients [137,140]. The major reason for the great attention to sotatercept is its ability to improve and reverse vascular remodeling of the pulmonary artery wall and right ventricle in PAH. Presently, a phase III open-label extension study is ongoing, further evaluating the safety and efficacy of sotatercept for the long term [82]. It is believed that this breakthrough therapy for PAH will promisingly achieve clinical value and benefit patients.

### 4.4. Seralutinib

Seralutinib acts as a kinase inhibitor targeting platelet-derived growth factor receptor (PDGFR)/colony-stimulating factor 1 receptor (CSF1R)/c-KIT, designed to be administered by inhalation for PAH treatment. These cytokines accumulate in vascular lesions, causing inflammation and abnormal proliferation in lung blood vessels. By inhibiting these inflammatory factors, seralutinib can effectively improve PAH, which has been confirmed in preclinical studies and clinical trials [129]. It is worth mentioning that seralutinib has a better curative effect than imatinib, and it is more suitable for inhalation delivery with low oral bioavailability [141]. Inhalants are more beneficial for treating lung diseases and can reduce systemic side effects, suggesting the feasibility of PAH clinical application. Given the clinical significance in the completed phase II trials, seralutinib is currently at the stage of phase III clinical trials.

Preclinical trials have proven the significant efficiency of inhaled seralutinib in PAH, including the reversion of pulmonary vascular remodeling and right ventricular hypertrophy with improvement in cardiopulmonary hemodynamic parameters. Inhaled seralutinib can inhibit the activity and signal transduction of PDGFRα/β, CSF1R, and c-KIT while increasing BMPR2, especially improving the inflammation around pulmonary vascular lesions [129,141]. Recently, the completed TORREY trial (NCT04456998) in patients with the treatment of inhaled seralutinib has met the primary endpoint. In this phase II, randomized, multicenter, double-blind, placebo-controlled study, PAH patients were treated with seralutinib by generic dry powder inhaler twice daily for 2 weeks at a dose of 60 mg or 90 mg. The feasibility of this treatment was demonstrated by the significant improvement in pulmonary vascular resistance after 24 weeks with slight adverse reactions in PAH patients [142]. To further evaluate the efficacy and safety of oral inhalation seralutinib for PAH therapy, a Phase III clinical study (NCT05934526) has already started, gaining the expectation of meaningful clinical application.

## 5. Conclusions and Perspectives

In recent years, the research and development of PAH drugs has been gradually deepened and become more extensive. Multiple new therapeutic targets for PAH, including inhibition of vascular remodeling, improvement in right heart function, inhibition of cell proliferation, regulation of gene expression, anti-inflammatory, and other mechanisms, have been confirmed by animal models and some drugs have begun clinical trials. The significant effect of inflammation and immunity in PAH pathogenesis has received considerable attention. A growing body of research has implicated that immune cells and inflammatory cytokines perform a noticeable function in PAH deterioration. Despite the inflammatory and immune mechanisms of PAH having not yet been exactly elucidated, a deeper understanding is still worth continuing. Recognizing the importance of inflammation as a pathological driving factor provides inspiration and direction for the development of new therapeutic methods for PAH.

Over the past decades, various classical therapeutic drugs for PAH have been widely used in clinical practice, but the poor prognosis of PAH patients has not been effectively improved. Most existing clinical drugs for treating PAH are vasodilators, which cannot improve the underlying pathogenesis of pulmonary vascular dysfunction. Targeted therapies related to inflammation, as a new pathophysiological intervention, have shown promising therapeutic potential for treating PAH. These potential drugs may perform independent or complementary therapy through the improvement of vascular remodeling. Recently, many existing or novel immunosuppressants have been demonstrated to perform encouraging results in preclinical or clinical trials. However, presently still no immunomodulator is approved for PAH therapy. Given the pivotal impact of inflammation and immunity for this disease, it is worth considering that anti-inflammation and immunotherapy have the potential to become new strategies for the clinical treatment of PAH.

In conclusion, it is of great significance to understand and explore the functions of immunity and inflammation in PAH for realizing efficient clinical translations of related immunotherapy. With processive research and collaborative effort, these potential therapies are expected to perform promising improvements for PAH and clinically benefit patients from disease therapy and life quality.

## Figures and Tables

**Figure 1 ijms-25-08427-f001:**
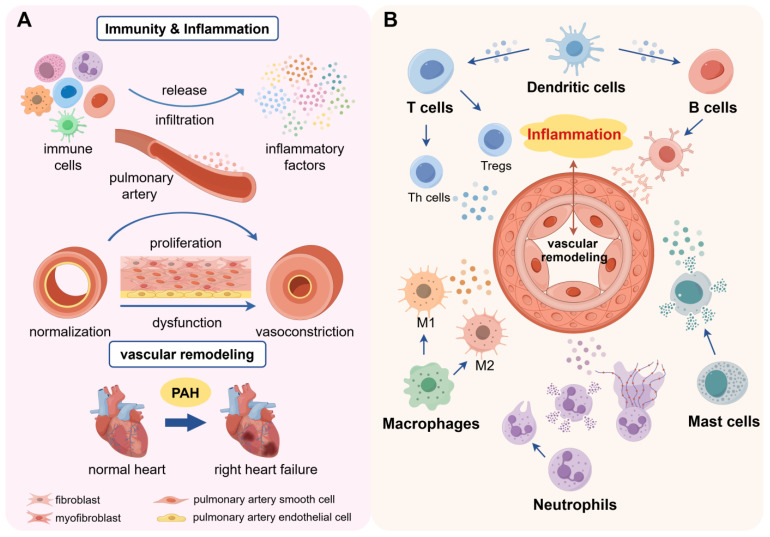
The function of immunity and inflammation in the pathogenesis of PAH: (**A**) Pathological development of pulmonary hypertension mediated by inflammation and immunity. The inflammatory environment around the pulmonary blood vessels is changed, manifested by the infiltration of immune cells and the production and recruitment of inflammatory factors. These reasons contribute to the transformation of PASMCs, PAECs, and PAFs to proliferative and proinflammatory phenotypes, which in turn generate pulmonary artery constriction and remodeling. In this state, PASMCs change from contractile to synthetic phenotypes through dedifferentiation with improved ability of proliferation and migration. Meanwhile, the abnormal condition causes damage and dysfunction of PAECs, exhibiting anti-apoptosis and excessive proliferation. It also stimulates the generation of myofibroblasts (MFs), carrying the characteristics of smooth muscle cells, derived from the differentiation of fibroblasts. MFs can migrate from adventitia to intima, promoting the thickening of blood vessel walls. Therefore, the involvement of inflammation exacerbates pulmonary vascular dysfunction, leading to disease progression and eventually right heart failure. (**B**) Pulmonary vascular remodeling is correlated with immune cells and inflammatory factors. There is infiltration of immune cells in pulmonary vascular lesions of PAH, including T lymphocytes, B lymphocytes, macrophages, mast cells, dendritic cells, and neutrophils. Meantime, relevant cytokines, chemokines, and antibodies, released or recruited around pulmonary vessels, also participate in the progression of vascular remodeling. See Table 2 for a more detailed explanation.

**Figure 2 ijms-25-08427-f002:**
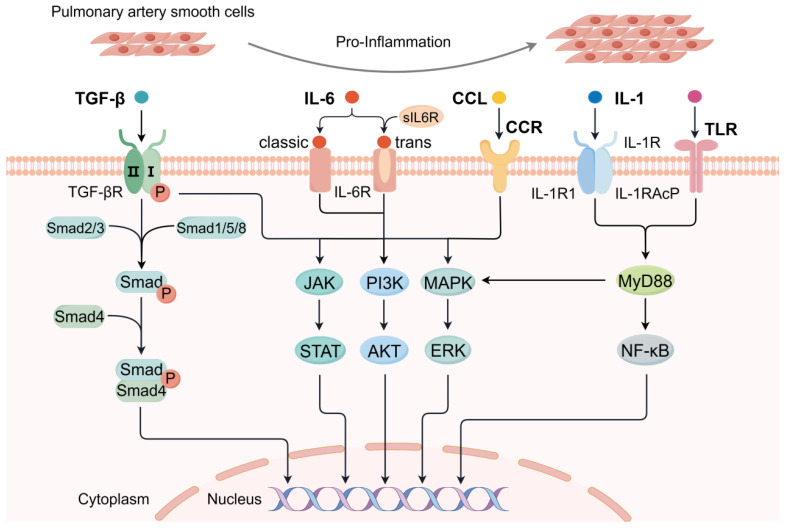
Molecular pathway mechanisms of cytokines and chemokines associated with inflammation for the progression of pulmonary hypertension. TGF-β signaling pathway is divided into classical (Smad-dependent) and nonclassical (Smad-independent) pathways. After the ligands bind to the complex of type I and type II receptors, type II receptors induce phosphorylation of type I receptors, causing the recruitment of Smad. The signaling transduction is that Smad1/5/8 or Smad 2/3 translocates into the nucleus by forming a complex with Smad4. In the noncanonical way, TGF-β receptor complexes activate signals through other factors. IL-6 combines IL-6R (classic) or IL6R with soluble IL6R (trans) to mediate the activation of downstream pathways. Through the formation of the IL-1 and IL1R (IL1RI and IL-1R accessory protein) complexes, related pathways become active after interacting with MyD88. Certain chemokines participate in the relevant signal transduction by binding to corresponding receptors, such as CCL2/CCR2. Also, the involvement of TLR is regarded as an important factor. The downstream signaling pathways consisting of MAPK and Nf-κB contribute to life processes by regulating gene expression. The effects of these immune and inflammatory pathways on the pathogenesis of PH are of significance for targeted therapy and disease treatment.

**Figure 3 ijms-25-08427-f003:**
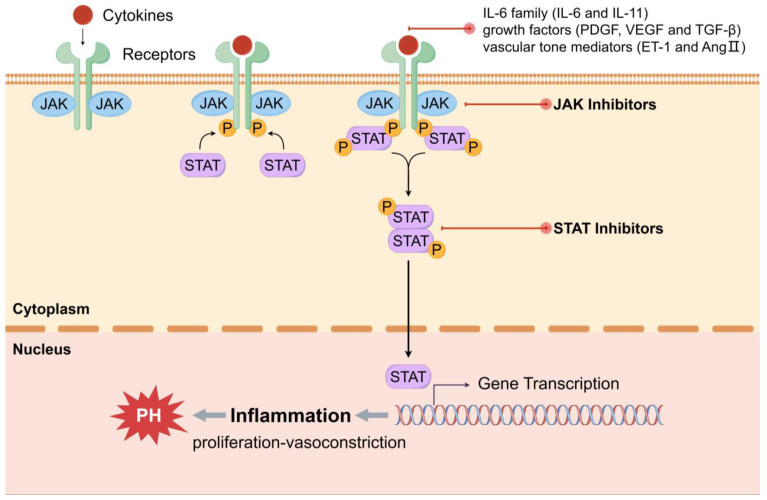
Effect of JAK-STAT signaling on the development of pulmonary hypertension. As a significant inflammatory pathway, JAK-STAT signaling is overactivated in PH, contributing to the pathogenesis of this disease. JAK families have four proteins comprising JAK1, JAK2, JAK3, and Tyk2, and human STAT families include seven subtypes, which are STAT1, 2, 3, 4, and 5A, 5B, and 6. The activators of JAK-STAT signaling in PH are composed of cytokines of the IL-6 family (IL-6 and IL-11), growth factors (PDGF, VEGF, and TGF-β), and vascular tone mediators (ET-1 and AngII). Once certain cytokines bind to the receptor on cell surface, JAK phosphorylates automatically at tyrosine residues, which in turn phosphorylates STAT and dimerizes with it. Subsequently, they enter the nucleus, bind to specific promoters, and transcribe genes correlated with cell proliferation, differentiation, and apoptosis. Targeting JAK, STAT, and related cytokines and receptors, JAK-STAT signaling might be a promising therapy for the clinical application and treatment of PH.

**Figure 4 ijms-25-08427-f004:**
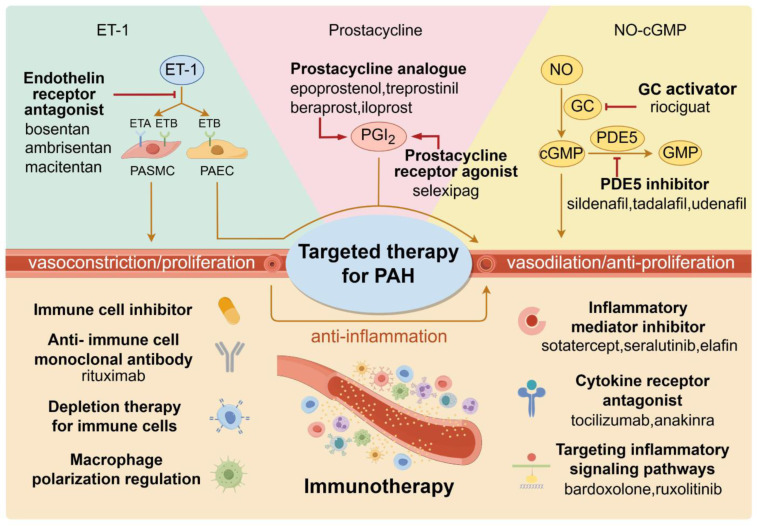
Clinical treatments and immunotherapy for targeting PAH. At present, the main clinical therapy of PAH is through ET-1, prostacyline, and NO-cyclic guanosine phosphate (cGMP) pathways, mediating vasodilation and resisting proliferation to achieve therapeutic attainment. ET-1 plays a vasoconstriction role by binding endothelin receptor type A and B (ETA and ETB) on PASMCs, which are inhibited by endothelin receptor antagonists. Because of the vasodilating function of prostacycline (PGI2), prostacycline analogs and prostacycline receptor agonists have been developed to alleviate PAH symptoms. The relaxation effect of NO in PASMC is realized by increasing the level of cGMP through activating guanylate cyclase (GC). On account of inhibiting the degradation of cGMP via phosphodiesterase type 5 (PDE5), PDE5 inhibitors have been applied clinically in PAH patients. Immunotherapy shows a promising prospect for PAH treatment, with the vital impact of inflammation in pulmonary vascular dysfunction gradually revealed. Over the years, diverse approaches have been developed to effectively treat this disease, based on the immune cells, inflammatory cytokines, and related signaling pathways. These therapies mainly include inhibitors for immune cells and inflammatory mediators, cytokine receptor antagonists, anti-immune cell monoclonal antibodies, depletion therapy for immune cells, macrophage polarization regulation, and targeting inflammatory pathways like JAK-STAT signaling.

**Table 1 ijms-25-08427-t001:** Inflammatory mechanism of pulmonary vascular cells in pulmonary hypertension.

Type	Position	Mechanism	Expression
Endothelial cells	intima	Develop proinflammatory phenotypes. Highly express and secret proinflammatory cytokines, chemokines, and adhesion molecules. Mediate inflammatory signaling transduction.	IL-1α, IL-6, IL-8, IL-12, CCL2, TNF-α, ICAM-1, VCAM-1, E-selectin, P-selectin.
Smooth muscle cells	tunica media		IL-6, TGF-β, TNF-α, CXCL8, ICAM-1, VCAM-1, P-selectin.
Fibroblasts (Myofibroblasts)	adventitia		IL-1β, IL-6, TGF-β, CCL2, CXCL12, CCR7, CXCR4, VCAM-1.

Abbreviations: IL: interleukin; TGF: transforming growth factor; TNF: tumor necrosis factor; CCL: C-C chemokine ligand; CXCL: C-X-C chemokine ligand; CCR: C-C receptor; CXCR: C-X-C receptor; ICAM-1: intercellular adhesion molecule-1; VCAM-1: vascular cellular adhesion molecule.

**Table 2 ijms-25-08427-t002:** The mechanism and strategy of immune cells and their secreted inflammatory cytokines in pulmonary hypertension.

Immune Cells	Inflammatory Cytokines	Mechanism	Drug Strategy
T lymphocytes	IL-1, IL-2, IL-3, IL-4, IL-5, IL-6, IL-9, IL-10, IL-13, IL-14, IL-17, IL-21, IL-22, IL-23, IL-35, TGF-β, IFN-γ, CCL2, CCL3, CCL4	T cell subtypes (helper T cells, cytotoxic T cells, and regulatory T cells) contribute to the pathogenesis of PAH. Immune response associated with CD4+ T cells, crucially acts in the development of inflammatory vascular lesions in PAH. Regulatory T cells inhibit PAH progression by exerting negative regulation on T-cell-mediated immune responses.	T cell signal transduction inhibitor anti-T cell monoclonal antibody
B lymphocytes	IL-2, IL-4, IL-6, IL-35, TGF-β, IFN-γ, CCL17, CCL22	B cells involved in the induction and progression of PAH. B cells and antibodies they produce have a potential impact on PAH, especially combined with autoimmunity.	anti-B cell monoclonal antibody B cell depletion therapy
macrophages	IL-1, IL-6, IL-8, IL-10, IL-12, IL-15, IL-17, IL-18, IL-23, TNF-α, TGF-β, IFN-γ, CCL1, CCL2, CCL5, CCL17, CCL18, CCL22, CCL24, CXCL9, CXCL13, CXCL16, M-CSF, GM-CSF, VEGF, PDGF	The recruitment of macrophages in pulmonary vascular lesions. The polarization phenotype of macrophages has a causal relationship with PAH aggravation. Inflammatory factors secreted by M1 and M2 macrophages contribute to the pathogenesis of PAH.	inhibitor blocking macrophage recruitment macrophage activation inhibitor macrophage phenotypic transformation regulator
mast cells	IL-1β, IL-2, IL-3, IL-4, IL-5, IL-6, IL-9, IL-10, IL-11, IL-13, IL-16, IL-33, TNF-α, TGF-β, CCL1, CCL2, CCL3, CCL4, CCL5, CCL7, CCL9, CCL10, CCL17, CXCL2, CXCL8, CXCL10, G-CSF, NGF, FGF, SCF, PDGF, VEGF	Mast cells infiltrate and degranulate in lung tissue and participate in pulmonary vascular dysfunction, promoting the progression of PAH. The mediators (proteases and lipid mediators) are released and activate degranulation of mast cells interacting with vascular cells to promote pulmonary vascular remodeling.	mast cell stabilizer or inhibitor inhibitor of related proteases and lipid mediators
dendritic cells	IL-1, IL-4, IL-6, IL-8, IL-10, IL-12, IL-15, IL-16, IL-17, IL-18, IL-23, IFN, TNF-α, CCL2, CCL3, CCL4, CCL5, CXCL9, CXCL10	Dendritic cells interact with T cells and B cells by producing and activating certain cytokines, contributing to disease pathogenesis of PAH.	targeting dendritic cell (T-DC) vaccine technology modulator of related cytokines
neutrophils	IL-1, IL-4, IL-6, IL-8, IL-10, TNF-α, TGF-β, IFN-γ, CCL2, CCL3, CCL5, CXCL1, CXCL8, CXCL10, G-CSF, VEGF	Neutrophils, neutrophil elastase (NE), and neutrophil extracellular traps (NETs) are interrelated with vascular remodeling progression and PAH pathogenesis.	neutrophil elastase inhibitor Immunotherapy targeting neutrophils

Abbreviations: IL: interleukin; TGF: transforming growth factor; TNF: tumor necrosis factor; IFN: interferon; CCL: C-C chemokine ligand; CXCL: C-X-C chemokine ligand; M-CSF: macrophage colony-stimulating factor; G-CSF: granulocyte colony-stimulating factor; GM-CSF: granulocyte-macrophage colony-stimulating factor; NGF: nerve growth factor; FGF: fibroblast growth factor; SCF: stem cell factor; VEGF: vascular endothelial growth factor; PDGF: platelet-derived growth factor.

**Table 3 ijms-25-08427-t003:** Preclinical studies in pulmonary hypertension targeting JAK-STAT signaling pathway.

Drug	Target	Mechanism	Disease	Model	Status
Ruxolitinib [115,116]	JAK1, JAK2	JAK1 and JAK2 inhibitor	PAH	Human PASMCs of IPAH patients, rats with MCT-induced, mice with Hox-induced	Preclinical models of PAH
			CTEPH	Rats induced by repeated embolization of the pulmonary artery with partially biodegradable 180 ± 30 μm alginate microspheres	Animal models of CTEPH
Fedratinib (TG-101348) [112]	JAK2	JAK2 inhibitor	PAH	Human PASMCs with hypoxia-induced	In vitro
Cucurbitacin I (JSI-124) [111]	JAK2, STAT3	JAK2 inhibitor	PH	Human PAECs and PASMCs of IPF patients, rats with bleomycin-induced	Preclinical models of PH
Magnolol [118]	JAK2	JAK2 inhibitor	PAH	Rats with hypoxia-induced, rat heart-derived H9c2 cells with hypoxia-induced	Preclinical models of PAH

**Table 4 ijms-25-08427-t004:** Clinical trials in pulmonary hypertension for immunotherapy.

Drug	Target	Mechanism	Disease	Phase	ClinicalTrials.gov. Identifier	Outcome
Sotatercept [82]	ActRIIA	IgG-ActRIIA fusion protein and TGF-β inhibition	PAH	III	NCT03496207	Completed
Seralutinib [129]	CSF1R, c-Kit, PDGFR	kinase inhibitor	PAH	III	NCT05934526	Recruiting
Tocilizumab [130]	IL-6R	IL-6 receptor antagonist	PAH	II	NCT02676947	Completed
Anakinra [100]	IL-1R	IL-1 receptor antagonist	PAH	Ib/II	NCT03057028	Completed
Rituximab [35]	CD20+ B cells	anti-CD20 chimeric monoclonal antibody	SSc-PAH	II	NCT01086540	Completed
Satralizumab [131]	IL-6R	anti-IL-6 receptor antibody	PAH	II	NCT05679570	Recruiting

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
