# Peer review of "Immunotherapy for Pulmonary Arterial Hypertension: From the Pathogenesis to Clinical Management"

_ijms, 2024, doi:10.3390/ijms25158427_

Round 1
Reviewer 1 Report
Comments and Suggestions for Authors
This manuscript addresses the important issue of inflammation and immunity in the development of pulmonary hypertension. Overall, this submission provides a nice overview of immune function in the development of pulmonary arterial hypertension, gene pathways, and clinical insight. However, there are a few topics presently lacking that could enhance the presentation of this discussion.
Major
1. At present, Table 1 and Table 2 seem to be merged together. It would be beneficial to provide a clearer delineation between these two items to improve clarity for the information contained within these tables.
2. The authors provide an excellent overview of the transcription pathways in pulmonary vascular cells which can modulate pulmonary vascular remodeling and pulmonary hypertension. However, little attention is given for immune function/inflammation for effects that are not mediated by transcription.
3. The lack of discussion for the role of myofibroblasts in the development of pulmonary hypertension should be addressed.
Minor
1. Line 532: It appears that there are some words missing prior to reference 114 to complete the thought.
Comments on the Quality of English LanguageEditorial Comments
1. First author last name should be capitalized on title page.
2. Line 279: H2O2 is missing subscripts.
Reviewer 2 Report
Comments and Suggestions for Authors
In a very interesting review, the authors presented the possibilities of immunological treatment of patients with PAH. I think I managed to describe the current and future directions well.
I only have minor comments on the work:
- in table 1 - there is information about, for example, the use of the antibody, but perhaps it would be better to add information about the nature of a given compound in each row - antibody, inhibitor, or small-molecule inhibitor
- in Figure 4. Clinical treatments and immunotherapy for targeting PAH - the phrase GC agonist appears. In relation to a roceptor, we can talk about a ligand: agonist, antagonist, inverse agonist, etc. while in relation to an enzyme it is an inhibitor, activator or modulator. In such a situation, riociguat is a modulator (it enhances enzyme stimulation in the presence of NO), unlike vericiguat - an activator. thus Soluble Guanylyl Cyclase Activators-Promising Therapeutic Option in the Pharmacotherapy of Heart Failure and Pulmonary Hypertension. Overall, Current Modulation of Guanylate Cyclase Pathway Activity due to Mechanism and Clinical Implications is a very interesting option, but it only applies to the enzyme described in the past as soluble, and generally, in relation to PAH, it does not apply to forms of membrane receptors.
Reviewer 3 Report
Comments and Suggestions for Authors
See the attachment

Needs Language editing. I did some highlighting in the manuscript but can´t upload the revised version.
Round 2
Reviewer 3 Report
Comments and Suggestions for Authors
Thank you for considering each of my comments and responding meaningfully to them. The writing and overall organization of the manuscript have been improved significantly.